# The Effect of Healing Perception on the Visitors’ Place Attachment and Their Loyalty toward a Metropolitan Park—Under the Aspect of Environmental Design

**DOI:** 10.3390/ijerph19127060

**Published:** 2022-06-09

**Authors:** Heng Zhang, Nam Nguyen-Dinh, Hazreena Hussein, Hong-Wei Ho

**Affiliations:** 1Department of Architecture, National Cheng Kung University, Tainan City 70101, Taiwan; n78077015@gs.ncku.edu.tw (N.N.-D.); homeway00@gmail.com (H.-W.H.); 2Centre of Sustainable Planning & Real Estate, Faculty of Built Environment, Universiti Malaya, Kuala Lumpur 50603, Malaysia; reenalambina@um.edu.my

**Keywords:** environmental design, place attachment, healing perception, place loyalty, structural equation modeling (SEM), full mediation effect

## Abstract

Healing perception is considered to increase visitors’ place attachment and loyalty. This research employed structural equation modeling (SEM) to examine the structural relationship between healing perception, place attachment, environmental design, and visitors’ loyalty to a place. The study investigated a metropolitan park in Gaoxiong, Taiwan, and collected 431 valid questionnaires on the site. The results showed that the environmental design affected the human perception of healing and place attachment, which substantially affected the visitors’ loyalty toward the place. The healing perception powerfully impacted loyalty (0.76), which contained an indirect effect through place attachment and enhanced the direct impact of healing perception. Moreover, the environmental design had a capable direct effect (0.62) on visitors’ loyalty through two full mediation paths: healing perception and place attachment. The study sheds light on designing a healing park that could enhance visitors’ place attachment and strongly affect their loyalty to the park.

## 1. Introduction

With the development of industry and commerce, the pursuit of work achievements causes physical fatigue, psychological pressure, and negative emotions in life, leading to a decline in mental health and causing chronic diseases eventually. Therefore, to seek a place where people can breathe a little under the tight pace of life in a big city is the key to balancing mental health [1]. Considering the elements that affect human psychological feelings and promote mental health and wellbeing, parks can be turned into a public leisure area with healing effects [2]. This is the reason why people often take time to visit parks—an open outdoor area in the city that they can easily reach.

In cites, Metropolitan Park is considered a suitable choice with a particular scale and rich in natural elements, which is often planned in urban areas or suburbs with high traffic accessibility and a specific scale. They have ecological conservation functions, environmental beautification, and leisure and recreation, enhancing physical and mental health to improve quality of life [3]. In other words, metropolitan parks can also provide disaster prevention, education, leisure and recreation, and access to the natural environment in terms of valuable functions [4,5,6]. Thus, this study surveyed the Gaoxiong Weiwuying Metropolitan Park to shed light on the influence of the environmental design of a park on the satisfaction of emotional and functional needs, healing environment perception, and visitors’ loyalty.

## 2. Literature Review

### 2.1. Environmental Design

The public open space of urban is a place for the people to socialize and interact [7]. In urban design research, public open space is defined as a park that provides greening and natural environment and has time control [4]. Park is used as a place that promotes health with two functions: “rest”, and “socialization”, it plays an ideal solution of environmental design to provide an outdoor field for the public [8]. Because the park is an integrated space of natural and man-made, its composition can include natural elements, use areas, structures, people, and other creatures [6], such as vegetation, soil, water, artworks, outdoor facilities, and buildings. It is necessary to consider passages, signs, entrances, exits, and safety during the planning and designing to achieve a compatible design that meets all users [9]. The manager and designers need to pay attention to designing the landscape that satisfies the three concepts: (1) Must be considered purposeful; (2) designed for the public; and (3) satisfy the needs of aesthetics and function [5].

In past studies, some scholars proposed to use the operable landscape elements in the land-laying plan as the basis and integrate them into six basic landscape elements: outdoor space, buildings, pavements, street furniture, artificial elements, and plants [10]. They are regarded as stimulation of the environment to the residents. In another article, the scholars further explored that the visual landscape elements positively influence the perception, emotion, and place attachment of the people [11]. In some cases, the scholars considered the visual level as the starting point to explore the visual aesthetic elements of the community, using architectural units, public spaces, natural elements, and human elements as the four aspects of planning and design elements [12].

Based on the composition of environmental design mentioned in the above documents, the environmental design defined in this study is composed of two parts, including the design operation elements of the site plan and the basic elements of the park design. According to the characteristics of the survey site, the environmental design is consolidated into six aspects: (1) recreational space, (2) building, (3) walks, (4) outdoor furniture, (5) planting, and (6) signboard.

### 2.2. Healing Perception

In the field of mental health, many scholars believe that mental fatigue will reduce the attention of people, produce negative emotions, increase the incidence of errors, and reduce the ability to solve problems as well [13]. However, they also pointed out that the natural environment can reduce stress and promote positive emotions and other positive benefits [14]. The restoration of functional resources and competence from depletion is known as the “healing effect” [15]. The theory of the healing effect was derived from two main theoretical models: the psychological evolution theory proposed by Ulrich and the other is the functional evolution theory based on the Kaplan couple.

According to Ulrich, the theory is focused on emotional and psychological reactions, emphasizing that the healing perception is based on emotion [16]. It is the first direct response when people interact with the environment; the healing effect of the environment will be improved through positive emotion so that stress can be reduced, rather than the direct attention can be restored [1,17].

Regarding Kaplan (1987), people will have special emotions for meaningful environmental information and establish human cognition in the natural environment and the functional results in the environment as well [18,19]. Then, human response toward the environment combines emotions, comprehensive judgment, and cognition [1]. In the “attention restoration theory” proposed by Kaplan and Kaplan (1989), autonomous attention and involuntary attention were regarded as direct attention and fascination, respectively [20]. If the humans use direct attention for a long time, it will cause “Direct Attention Fatigue” [21].

Although the two theories are not the same, there are still many common points. Many scholars used these theories to explore whether religious spaces and other fields have healing perceptions and then achieve the healing effects on human mental health [22]. Environmental designs can produce healing perceptions and have the potential factors to achieve healing effects [23,24,25]. Therefore, this study combined the two above theories to explore the relationship between the environment and the healing perception, in which the healing space was represented by four characteristics: (1) distance, (2) extension, (3) fascination, and (4) compatibility.

### 2.3. Place Attachment

“Place” is composed of physical forms, activities, and meanings [26]. The result of the interaction between human and place is called place attachment. In the 1970s and 1980s, the branch of human geography mentioned the idea of place, which profoundly impacted the concept of place and place attachment [27,28,29]. Some scholars believe that place attachment refers to the emotional and functional connection between the individual and the place through the interaction of emotion, knowledge, belief, and behavior [30]. The emotional/symbolic connection is often called place identity, whereas the functional requirements are called place dependence [31].

Place identity refers to the sense of attachment and belongingness to the place at the emotional level and is the individual emotional attachment to the place or the environment [32]. As merely, place identity is a personal sense of identity with a place, enabling individuals to distinguish this place from other places [11]. Place dependence refers to the functional connection between the individual and the place [33], when the individual feels essential to a particular place and can meet their functional needs and support their behavioral goals more than available alternatives [34].

Based on the theoretical literature above, place attachment composes two components, including place identification for emotional belonging and place dependence for functional needs, as discussed in many studies. Therefore, this study divided place attachment into two aspects: (1) emotional attachment and (2) functional attachment.

### 2.4. Loyalty

As a psychological factor in human behavior, loyalty is considered the measurement of subjective judgment of the unique possibility to engage in a particular behavior, which corresponds to the individual willingness to engage in a particular behavior and an index to measure future behavior [35]. The researchers discussed the four different loyalty stages for the representative model in loyalty research, including cognitive, emotional, intentional, and behavioral loyalty [36]. Cognitive loyalty is the product information that consumers are aware of/or obtain and is the weakest form of loyalty; the subsequent commitment is emotional loyalty. Emotional loyalty is the attitude of the consumer toward products, and the relationship between products is determined. If consumers have a good attitude toward the product, they will be emotionally loyal to it [37]. Intentional loyalty is the behavioral intention stage, while the final behavioral loyalty is the conversion of intention to action. Some scholars transferred the four loyalty stages into two aspects: attitude loyalty and behavior loyalty [38].

In marketing research, loyalty is defined as a sense of pursuit of people, products, or services promised to consumers, willingness to repurchase, and preference [39]. Once an individual believes that he is receiving the best service, his loyalty toward this service will of course be enhanced. He can evaluate the products of this service as the first choice the next time, or he can be willing to recommend the positive words about products to others [40].

In the research field of environmental behavior and tourism, many studies infirmed that behavioral intention is a major representation of loyalty [41]. They often evaluate loyalty by measuring perception, emotion, and satisfaction toward the environment and then predicting behavioral intentions of visitors in the future, such as willingness to revisit, recommend to others, and loyalty to similar attractions [42,43]. In this paper, loyalty is divided into two aspects: (1) attitude loyalty, and (2) behavior loyalty.

## 3. Hypothesis Development

### 3.1. Environmental Design and Healing Perception

The natural environment positively affects psychological recovery, but less attention has been paid to the importance of the physical environment that contributes to the healing potential [44]. The importance of environmental landscape can be divided into “cognition content”, “activity content”, and “well-being content” [45]. These contents are the sources of healing possibilities of the environmental landscape. The activities in the natural environment can provide visitors a positive impact on mental health and healing effects and reduce the pressure caused by urban life rather than in the urban environment [46].

Healing research emphasized the potential elements of natural outdoor environments such as trees, lakes, and mountains to reduce stress, fatigue, and improve mood. Moreover, according to the attention restoration theory [21], natural environment has the characteristics of replenishing energy and restoring attention. When observing the environment and aesthetic characteristics, it is considered that the recovery of attention is a kind of healing. When individuals are obsessed with something, they do not need to try to guide their attention, so the attention will not become fatigued but can attract attention to power recovery from fatigue.

**Hypothesis** **1** **(H1)**.*Environmental design positively influences healing perception*.

### 3.2. Healing Perception and Place Attachment

Past studies, which link place and healing environment, proposed solutions related to place attachment and healing perception and verified the positive predictive power of place identity, place dependence, and healing perception. These findings showed the relevance of experience and personal attachment to the healing environment [47]. If an individual has a positive emotional attachment to a place, including his preference and familiarity with the place, the attached place also has healing potential [48].

There have been many related studies on place attachment, and in recent years it has been discussed in the study of leisure and recreation, for example the relationship between place attachment, satisfaction, and loyalty of tourists to recreational areas. Some scholars argued that the natural environment could promote human mental health and achieve healing better than the urban environment [49,50].

**Hypothesis** **2** **(H2)**.*Healing perception positively influences place attachment*.

### 3.3. Environmental Design and Place Attachment

The definition of “place” includes the physical environment and the social construction of local consciousness [51]. The perspective of the place originated from the environmental landscape, and the hidden meaning of the environmental landscape can be discovered from the place dependence [19].

The physical environment is one of the critical aspects to promote the sense of attachment. However, in most cases, the sense of attachment is intertwined with the natural environment and social interaction [51]. Therefore, it is necessary to study further the role of the physical and natural environment in the sense of attachment [52]. The concept of the sense of place includes three elements: location, landscape, and personal involvement [53].

The location and landscape include volumetric forms of buildings, movement lines, planting, green spaces, paving, and street furniture [11]. Through planning and design, the environment can be formed as an identity space where people can move, give meaning, and be emotionally connected with space. Some scholars believed that when the environment of the place satisfies the needs of people, it becomes more meaningful for them and causes people to have a dependent feeling, a sense of attachment, and hope for the further development of the place [11,19]. On the other hands, when the individual blends into the place, more profound sense of attachment and belongingness to the place and a clearer feeling of the importance of the place develop which confirm the human has an attachment with the place [54].

**Hypothesis** **3** **(H3)**.*Environmental design positively influences place attachment*.

### 3.4. Environmental Design, Place Attachment, Healing Perception, and Loyalty

The urban landscape environment comprises natural and man-made environments with vegetation, water, lighting, outdoor furniture, pavement, building, artwork, and even signboards that can significantly affect human psychology, in which they can touch human’s healing perception that will also promote their mental health, reduce arousal, improve mood and cognition, thus, promoting the processing of mental health restoration positively [11,46,55].

Natural elements positively impact visitors in this relationship, in which man-made elements or layout planning can also affect satisfaction [36,56]. However, many scholars rate tourist loyalty with high confidence because it is a more accurate indicator of actual behavior than satisfaction [57]. Kim and Brown (2012) found a significant relationship between exposure to nature, revisiting, and recommendation [58]. If the characteristics of the place are sufficiently attractive to the visitors, they will encourage them to go [56]. Through the impression of the layout of the seating space, artwork, performers, street accessibility, and trees in urban public spaces, the willingness and attachment toward the place of visitors can be booted. The visitors choose to visit a place because they can achieve a specific goal or activity demand or its symbolic meaning. Therefore, attachment is an essential indicator of tourist loyalty. Attachment can affect the perceptions, thoughts, and feelings of visitors. Increasing the knowledge and emotional connection of a place may increase the possibility of positive evaluation and loyalty to the place in humans [59].

On the other hand, when referring to recreation and leisure in modern society, Hurd, Anderson [60] pointed out that a world is unfathomable to other parts without recreation and leisure. Similar to other scholars, he infirmed parks, open space, sports, and traveling play an important role in enhancing recreation and leisure in nature that can touch the lives of almost everyone. In the study of the relationship between the orientation of place attachment and the loyalty of tourists, the scholars found that functional attachment and emotional attachment are factors that predict the intention of loyalty to the park and determine their behavioral intentions. The higher the sense of attachment of tourists to places, the higher their loyalty, and vice versa [54]. Loyal visitors are especially significant, making other users willing to pay more attention by making a positive word of mouth and sharing an intention of visiting in the future or giving their recommendations to others. Those interactions could create publicity effects [61]. Therefore, if the function of landscape space satisfies the visitors, it becomes an essential predictor of loyalty in the future.

**Hypothesis** **4** **(H4)**.*Healing perception positively influences loyalty*.

**Hypothesis** **5** **(H5)**.*Place attachment positively influences loyalty*.

**Hypothesis** **6** **(H6)**.*Healing perception has a mediating influence on the relationship between environmental design and place attachment*.

**Hypothesis** **7** **(H7)**.*Place attachment has a mediating influence on the relationship between healing perception and loyalty*.

**Hypothesis** **8** **(H8)**.*Healing perception and place attachment mediate the relationship between environmental design and loyalty*.

Based on the above theoretical literature, the study argued there is a causal relationship existing between the four dimensions: urban park environmental design, place attachment, healing perception, and loyalty of visitors. Hence, the study aimed to explore the correlation research and discussion of the four dimensions to shed light on this cause relationship with six effect hypotheses as below (Figure 1):

## 4. Methodology

### 4.1. Study Site

The Greater Gaoxiong area is a big city in Taiwan where industrial zones are located densely. The living of citizens is affected by the high pressure of work, high-density construction, and traffic, so that the need of a public open space is necessary where they can visit, leisure, and relax. However, it is not only in the natural environment that the healing effect can be achieved, but the beauty and other factors, such as pavement, gazebo, outdoor furniture, and activity spaces, also play crucial impacts. Therefore, the study selected Gaoxiong Weiwuying Metropolitan Park, Fongshan district, as the scope of research, which has natural landscape, artistic and cultural conditions for residents and visitors, as the field of discussion to explore the relevance of environmental design, place attachment, healing perception, and loyalty (Figure 2).

After 20 years of efforts by civil groups that advocated environmental protection and arts, the Weiwuying Metropolitan Park was established in 2010 and is located in the Fongshan district, south of the Gaoxiong City. The City Government decided to investigate some cultural buildings in this park, such as the National Gaoxiong Center for the Arts, the ecological zone, and the cultural activity area. The architecture and layout of these buildings play as highlight points for the park. Especially the Center for the Arts was built to satisfy the public’s need to enjoy art and culture, including the opera house, concert hall, playhouse, and recital hall. With the diversity of the vegetation types, the ecological zone of the park is considered the ideal living habitat for birds. In addition, a large grassland in the park is also very loved by the public as a green space to rest and recreate mental health after the pressure of the day-life.

### 4.2. Data Collection

The study used a quantitative research approach and conducted a face-to-face survey in Gaoxiong Weiwuying Metropolitan Park to collect data. The individual visitors in the field were considered as the sampling unit, and the sampling number was determined based on the sampling error. According to Naing, Winn, and Rusli (2006), the estimation formula of population sample number is finite [62]. The total population number of 2,778,375, the confidence interval of 95%, and the allowable error value of 5% are taken into calculation. Hence, the number of samples required for this study is 384. The sampling method adopted the convenient sampling method of the non-random sampling method. Based on the above review of the theoretical literature, the researchers built the questionnaires, distributed them to the visitors, and then collected them immediately after finishing the on-site answer.

In several two-person groups, the survey team went to the activity and sitting areas in the park to approach the potential participants. The surveyors introduced themselves, explained the purpose of the research, and requested participation in the questionnaire. A souvenir was given after the participants completed the questionnaire designed for this study and presented in Mandarin Chinese for local participants’ self-reporting. The visitors in the park were friendly toward the research, which is unintrusive to their privacy, and were willing to share their attitudes and feelings. The recruiting of participants was smooth.

### 4.3. Measurement of Variables

The study suggested fifty-three questions to measure the environmental psychology of visitors, which represent the four research constructs: the environmental design, place attachment, healing perception, and loyalty. The environmental design was described by twenty-three items (A01 to A23) within six factors: recreational space, building, walkway, outdoor furniture, planting, and signboard (see Appendix A). Place attachment was described by ten items (B01 to B10), extracted in two factors: emotional attachment and functional attachment (see Appendix B). Healing perception was described by sixteen items (C01 to C16), extracted in four factors: distance, extension, fascination, and compatibility (see Appendix C). Finally, the loyalty was described by four items (E01 to E04), extracted into two factors: loyalty attitude and behavioral loyalty (see Appendix D). The measurement items are originally written in Chinese. The study applied the five-point Likert scale to score these questions, with 1 indicating very disagree, 2 indicating disagree, 3 indicating ordinary, 4 indicating agree, and 5 indicating very agree.

#### 4.3.1. Item Analysis and Reliability Analysis

The main project analysis methods are correlation analysis, internal consistency, and benchmarking. Reliability refers to the degree of consistency of measurement. Cronbach’s α value is used as the basis for the reliability judgment, and its value is between 0 and 1. The higher the Cronbach’s α of the overall scale, the more reliable it is [56].

#### 4.3.2. Testing the Validity of the Measurement Models

The validity of the measurement model is evaluated to test whether the constructs and indicator variables can reflect the underlying constructs. Therefore, Confirmatory Factor Analysis (CFA) can be used to evaluate the fit and construct validity of the hypothetical model, including Chi-square/d.f., GFI, AGFI, CFI, NFI, RMR, and RMSEA [56,63].

Moreover, the convergent validity and discriminant validity must be tested. The convergent validity must meet the following conditions at the same time: (1) The factor load value of the question item must exceed 0.7 and be significant at the *t*-test; (2) construct reliability must be greater than 0.6; (3) the average variance extracted (AVE) of each dimension must be greater than 0.5 [63].

### 4.4. Structural Equation Modeling (SEM)

The estimation of the structural equation model will be the next stage, in which this approach evaluates the causal relationship between the four constructs. On the one hand, this method aims to calculate the direct effect values on the path between the constructs. On the other hand, the mediating effects will be tested to confirm whether the mediators exist in the model and estimate how strong they are.

## 5. Results

### 5.1. Profile of Sample

The sample survey was conducted from November 19 to November 25, and in the period time from 8 am to 5 pm. The survey team collected 449 responses from the 450 questionnaires distributed. About 18 uncompleted responses were eliminated, and the remaining valid responses were 431, which was higher than the threshold of 384 required samples [62]. The profile of sample is shown in Table 1.

### 5.2. Item and Reliability Analysis

Table 2 lists the reliability assessment results of the indicators representing constructs. According to the analysis, all constructs obtained a high value of Cronbach’s α range from 0.924 to 0.945. These indexes were higher than the threshold of 0.60 and informed that the data set is highly reliable for in-depth analysis [64].

### 5.3. Structural Equation Model Analysis

#### 5.3.1. Assessment of the Measurement Model

The stage of confirmatory factor analysis was conducted to test a measurement model composed of four dimensions: environmental design, place attachment, healing perception, and loyalty. Through the model fit evaluation mainly, the study aimed to compare the gap between the theoretical and actual results from various aspects. Under the support of the Amos software (version 22), the paper obtained the Goodness-of-fit index value (GFI) of 0.935, which shows that the model in this study can explain more than 90% of the data variation, and the mean square residual RMR of the model was 0.017, which is less than 0.05. The above results show that the proposed hypothetical model in this study is compatible with the actual observation data and can explain the relationship among the constructs.

According to the fit test results, the chi-square value (X^2^) of the chi-square test reached a significant level (X^2^ = 207.724, degrees of freedom is 71, X^2^/df = 2.926, *p* = 0.00). In terms of the absolute fit index, the model obtained the GFI value of 0.935, RMSEA value of 0.067, RMR value of 0.017, NFI value of 0.950, CFI value of 0.967, and the PNFI value of 0.741. The overall measurement model adaptation index has reached acceptable adaptation standards. Therefore, the validity of the measurement model and the collected data was confirmed.

In Table 3, the results showed the convergent validity of the four constructs, including environmental design, place attachment, healing perception, and loyalty.

The study evaluated the convergent validity of the measurement model, in which the factor load value, the extraction of average variance, and the composite reliability of the constructs will be tested. The factor loading values of predictors were between 0.696 and 0.925; the composite reliability (CR) values of the four constructs, including environmental design, place attachment, healing perception, and loyalty, were 0.889, 0.875, 0.887, and 0.887, respectively and exceed the threshold of 0.70; the average variance extraction (AVE) values of the four constructs were 0.574, 0.778, 0.663, and 0.797 approximately, greater than the threshold of 0.50.

In Table 4, the result exposed the discriminative validity of the measurement model. The correlation between items in different dimensions was low. The average extraction amount of variation for each facet was higher than the square of the correlation coefficient among constructs. Therefore, the measured model had discriminative validity.

#### 5.3.2. Assessment of the Structural Model

In this statement, the goodness-of-fit was assessed to test the validity of the structural model [63]. The analysis exposed that a good fit for the structural model was obtained (X^2^ = 210.167, df = 72, X^2^/df = 2.919, CFI = 0.966, GFI = 0.935, AGFI = 0.905, NFI = 0.950, RMSEA = 0.067, RMR = 0.017, PNFI = 0.751, *p*-value = 0.000), indicating that the validity of the structural model is good. The structural model of the four constructs is shown in Figure 3. The above results were all significant (*p* < 0.05), and the total variance explained was 68%.

#### 5.3.3. The Total Effects with Direct Effects and Mediating Effects in the Models

The analysis results accepted all hypotheses H1 to H8. They expressed five directly causal effects in the relationship between the four tested constructs and three indirect effects due to the influence of mediators. In particular, the direct effect of environmental design on the healing perception was the largest (0.74***), followed by healing perception on place attachment (0.66***) and healing perception on loyalty (0.56***). The causal relationship of environmental design on healing perception had a variance explained of 55%, whereas that of the environmental design, healing perception on place attachment had 63%.

Due to the mediator of healing perception, a large indirect effect existed from environmental design to place attachment (0.49***) significantly (z = 6.613, *p* = 0.000) [65], and was two times greater than the direct effect (0.17*). Hence, the total effect of environmental design to place attachment was large (0.66***) and emphasizes that the healing perception contributed substantially to the impact of environmental design to place attachment. The effect of environmental design and healing perception on place attachment was explained 64% of the variance.

Similarly, in the relationship among the healing perception, place attachment, and loyalty, the influence of healing perception on place attachment was the strongest (0.66***), followed by the effect on the loyalty (0.56***), while the influence of place attachment on the place loyalty was medium (0.31***). Moreover, the model also revealed that place attachment played a mediator in the path from healing perception to loyalty (0.20**) significantly (z = 3.116, *p* = 0.002) [65].

Although a direct effect from environmental design to place loyalty did not exist, environmental design affected loyalty through indirect effects of place attachment (0.31***) and healing perception (0.56***). The indirect effect in the structural relationship between environmental design, healing perception, and loyalty was calculated at 0.414*** and reached a significance level (z = 5.797, *p* = 0.000) [65]. In the structural relationship between environmental design, healing perception, place attachment, and loyalty, the indirect effect was obtained 0.204*** and indicated a significance level (z = 3.345, *p* = 0.001) [65]. Through the two sub-indirect effects above, the total indirect effect of the construct “environmental design” to “loyalty” reached a high value (0.62***), explaining 68% of the variance. Table 5 shows the total effects among the four constructs with direct and indirect effects.

## 6. Discussion

This research demonstrated the crucial role of environmental design through psychological mechanisms, such as healing perception and place attachment, to influence the visitors to stay longer and often come to the park to benefit from abiding there. The study results (Figure 3 and Table 5) showed that the healing perception is the essential construct of the model and functions as a predictor for place attachment and loyalty to the park. The healing perception further affects place attachment (0.66), subsequently the visitors’ loyalty. Though the healing perception impacts loyalty already by 0.56 directly, through place attachment, the effect enhanced to be 0.76, which is noticeably higher. The indirect impact through place attachment enhances the contribution of healing perception to loyalty by 35.7%.

Moreover, environmental design strongly influences the healing perception (0.74), affecting the place attachment (0.66 totally). Through visitors’ healing perception, the power of environmental design on place attachment is enhanced by 288.2%. When design elements, such as outdoor furniture and recreation space, are well planned, they enforce the healing perception of visitors. This mechanism enhances the influence of healing perception and place attachment to visitors’ loyalty. Though environment design has no direct impact on visitors’ loyalty, it provides an indirect effect, i.e., full mediation, on loyalty with a substantial extent of 0.62. In practice, planners and designers can utilize the relationship between environmental design, healing perception, and place attachment to create a cozy outdoor environment that improves human wellbeing. This principle is not only valid for the urban park, but should be promoted in any place which longs to have a healing effect on the visitors. All of the six indicators of environmental design in the model retain high factor loadings and can be employed as the critical design factors to conceive a healing environment.

This research result verifies the environment that space, facilities, and plantings, the itinerary is accompanied by spatial meaning to have an emotional connection to space [11,67]. Plantings such as lawns and trees have the most significant influence on park selection. Natural elements have a positive impact on tourists, and artificial elements, such as signboards, can also affect the perception of tourists. If the elements of the park are sufficiently attractive to tourists, it will encourage tourists to go there. When visitors go to a park, they can temporarily withdraw from the busy life, get respite from the pressure, retain spiritual release, and have a positive attachment to the place. The formation of a sense of place is related to the perception of healing, and personal preference for the environment will make the individual more dependent on the environment [26]. The same with the satisfaction of activity needs, it positively affects visitors’ functional attachment [33,34].

Attachment can affect the perceptions, thoughts, and feelings of visitors. Enhancing the emotional connection to a place may increase positive evaluation and loyalty to the place [68]. The specific attachment to the environment, which fulfills the needs of the visitors, will generate loyalty [57].

Though few literature argued that healing perception affects the loyalty directly, the study results indicated a direct effect of healing perception on park visitors’ loyalty with 0.56 and 0.76 totally, which is remarkable for psychologists and park designers. This result could be because the park provided an anticipated service [39] in the study of healing perception, thus strongly impacting visitors’ loyalty. The fact that the factor loading of functional attachment (0.92) is greater than that of emotional attachment (0.84) also agrees this presume. The mechanism that the study revealed can be used in the design and planning sector to create places with humanity and healing effect and should be introduced to the health care and wellbeing sector to enhance the healing perception by abiding in an appropriately designed public park.

## 7. Conclusions

According to the study results, the healing perception provided by a metropolitan park affects visitors’ place attachment strongly (0.66). Moreover, it heavily contributes to visitors’ loyalty directly and indirectly (totally 0.76). Due to the indirect influence through the path place attachment, healing perception enhances the impact on loyalty by 35.7%. Thus, the healing perception that a metropolitan park can deliver was noteworthy to visitors’ loyalty.

Moreover, the environmental design strongly affects the visitors’ healing perception (0.74). Simultaneously, the environmental design also intensely impacts the visitors’ place attachment (0.66). Via visitors’ healing perception, the environmental design enhanced its power on place attachment by 288.2%. This mechanism boosted the influence of healing perception and place attachment to visitors’ loyalty. Though the environment design has no direct impact on visitors’ loyalty, it still produced an indirect effect on loyalty strongly (0.62).

Constructing a healing metropolitan park that also meets the visitors’ functional and emotional needs can increase the place attachment of visitors. Healing perception together with place attachment of visitors contributes to their loyalty. An appropriate environmental design delivers visitors with high healing perception and place attachment, further improving visitors’ loyalty enormously.

## Figures and Tables

**Figure 1 ijerph-19-07060-f001:**
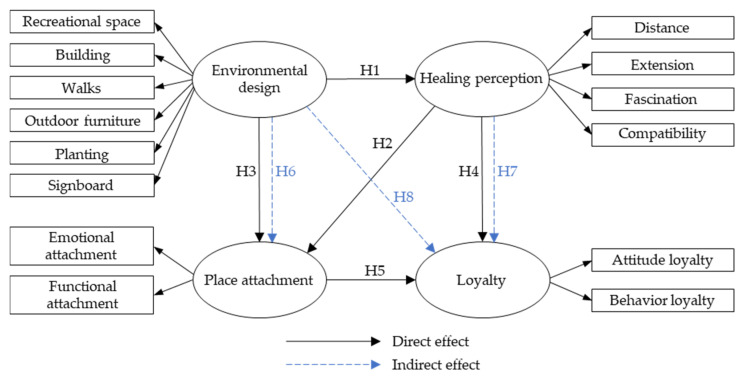
Proposed framework for the research.

**Figure 2 ijerph-19-07060-f002:**
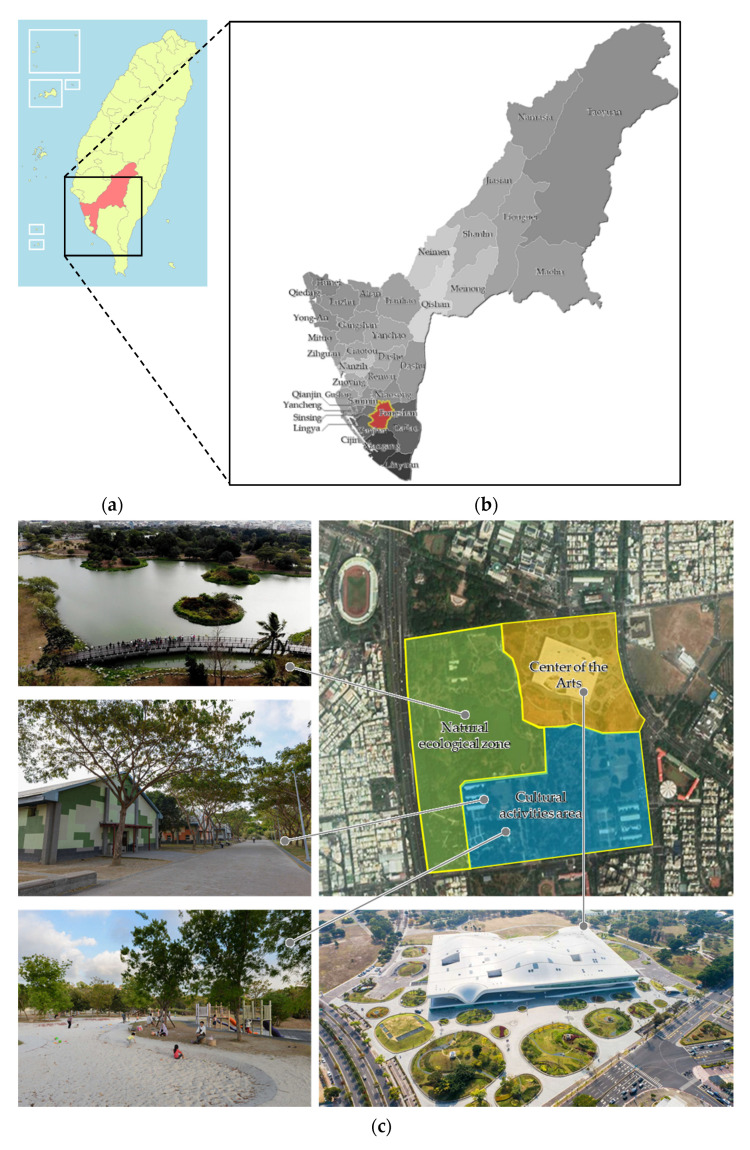
(**a**) Map of Taiwan; (**b**) Map of Gaoxiong; (**c**) Gaoxiong Weiwuying Metropolitan Park.

**Figure 3 ijerph-19-07060-f003:**
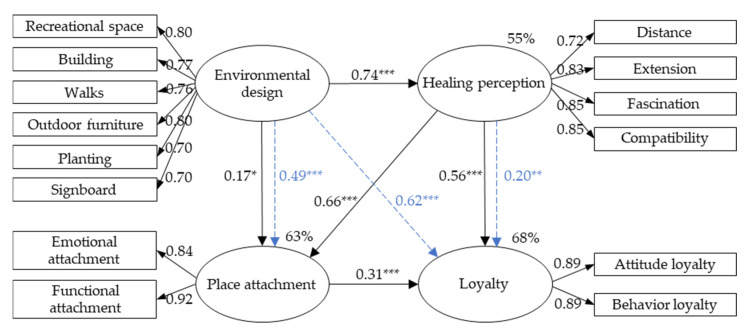
The suggested model of the relationships among environmental design, healing perception, place attachment, and place loyalty. The dashed lines indicate the indirect effects between the pair of constructs. * *p* < 0.05; ** *p* < 0.01; *** *p* < 0.001.

**Table 1 ijerph-19-07060-t001:** Demographic characteristics of the respondents.

Characteristics	Sample	Percent
Gender		
Male	209	48.5
Female	222	51.5
Age		
11–20	10	2.3
21–30	80	18.6
31–40	189	43.9
41–50	113	26.2
51–60	15	3.4
Above 61	24	5.6
Residence		
Local residence	184	42.7
From other local	247	57.3
Education		
Elementary	4	0.9
Secondary school	18	4.2
High school	83	19.3
Graduate	248	57.5
Postgraduate	78	18.1
Frequency		
under 2 times	124	28.8
2 times	106	24.6
3 times	92	21.3
4 times	88	20.4
above 5 times	21	4.9

**Table 2 ijerph-19-07060-t002:** Reliability assessment of the constructs.

Construct and Indicators	Mean	Std. Deviation
Environmental design (α = 0.945)		
Recreational space (A01 to A04)	3.80	0.663
Building (A05 to A08)	3.86	0.700
Walks (A09 to A11)	3.81	0.679
Outdoor furniture (A12 to A15)	3.56	0.688
Planting (A16 to A19)	3.92	0.755
Signboard (A20 to A23)	3.55	0.715
Place attachment (α = 0.942)		
Emotional attachment (B01 to B05)	3.71	0.737
Functional attachment (B06 to B10)	3.73	0.780
Healing perception (α = 0.945)		
Distance (C01 to C04)	3.90	0.715
Extension (C05 to C08)	3.76	0.650
Fascination (C09 to C12)	3.70	0.672
Compatibility (C13 to C16)	3.96	0.619
Loyalty (α = 0.924)		
Attitude loyalty (E01 to E02)	4.06	0.705
Behavior loyalty (E03 to E04)	4.11	0.696

**Table 3 ijerph-19-07060-t003:** Convergent validity of measurement model.

Construct/Indicator	Factor Loading (λ)	Reliability Coefficient (λ^2^)	Measurement Error (1 − λ^2^)	AVE ^(2)^	CR ^(3)^
Environmental design (ED)				0.574	0.889
Recreational space	0.800 ***	0.640	0.360		
Building	0.772 ***	0.596	0.404		
Walks	0.760 ***	0.578	0.422		
Outdoor furniture	0.804 ^(1)^	0.646	0.354		
Planting	0.705 ***	0.497	0.503		
Signboard	0.696 ***	0.484	0.516		
Place attachment (PA)				0.778	0.875
Emotional attachment	0.837 ***	0.701	0.299		
Functional attachment	0.925 ^(1)^	0.856	0.144		
Healing perception (HP)				0.663	0.887
Distance	0.718 ***	0.516	0.484		
Extension	0.829 ***	0.687	0.313		
Fascination	0.848 ^(1)^	0.719	0.281		
Compatibility	0.854 ***	0.729	0.271		
Loyalty (LO)				0.797	0.887
Attitude loyalty	0.894 ^(1)^	0.799	0.201		
Behavior loyalty	0.892 ***	0.796	0.204		

Note. *** *p* < 0.001. ^(1)^ Significance was not calculated because the unstandardized loading was set as 1.0 to fix construct variance. ^(2)^ Average variance extracted = (∑λ^2^)/*n* (where n is the number of items). ^(3)^ Composite reliability = (∑λ)^2^/[(∑λ)^2^ + (∑error)].

**Table 4 ijerph-19-07060-t004:** Discriminant validity of measurement model.

	MSV	MaxR(H)	PA	ED	HP	LO
Place attachment (PA)	0.619	0.892	0.882			
Environmental design (ED)	0.543	0.893	0.660 ***	0.757		
Healing perception (HP)	0.634	0.895	0.787 ***	0.737 ***	0.814	
Loyalty (LO)	0.634	0.887	0.745 ***	0.654 ***	0.796 ***	0.893

Note. *** *p* < 0.001.

**Table 5 ijerph-19-07060-t005:** The total effects among the four constructs and the mediating effect of healing perception and place attachment.

Path	Direct Effect	Indirect Effect	Total Effect
ED→HP	0.74 ***		0.74 ***
HP→PA	0.66 ***		0.66 ***
ED→PA	0.17 *	0.49 ***^(1)^	0.66 ***
HP→LO	0.56 ***	0.20 **^(2)^	0.76 ***
PA→LO	0.31 ***		0.31 ***
ED→LO		0.62 ***	0.62 ***
ED→HP→LO		0.414 ***^(3)^	0.414 ***
ED→HP→PA→LO		0.204 ***^(4)^	0.204 ***

Note. * *p* < 0.05; ** *p* < 0.01; *** *p* < 0.001. ED-Environmental design; HP-Healing perception; PA-Place attachment; LO-Loyalty. Based on Sobel test [66], the indirect effects were examined as below: ^(1)^ ED→HP→PA: (0.74 × 0.66) = 0.49, z = 6.613, *p* < 0.001. ^(2)^ HP→PA→LO: (0.66 × 0.31) = 0.20, z = 3.116, *p* < 0.01. ^(3)^ ED→HP→LO: (0.74 × 0.56) = 0.414, z = 5.797, *p* < 0.001. ^(4)^ ED→HP→PA→LO: [(0.74 × 0.66) + 0.17] × 0.31 = 0.204, z = 3.345, *p* < 0.001.

## Data Availability

The data presented in this study are available on request from the corresponding author.

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
