# Peer review of "The Effect of Healing Perception on the Visitors’ Place Attachment and Their Loyalty toward a Metropolitan Park—Under the Aspect of Environmental Design"

_ijerph, 2022, doi:10.3390/ijerph19127060_

Round 1

Reviewer 1 Report

This is a very interesting study. I appreciate the authors' efforts in conducting this research with valuable implications. The manuscript was well-written, and the research was nicely executed.

Here are several issues I recommend for revisions:

1. The structure of the front-end could be further improved for more clarity. Since you have included several sub-dimensions underneath each of the four main constructs: environmental design, healing perception, place attachment, and loyalty, it would be better to illustrate all the sub-dimensions either in the conceptual model (Figure 1) or create a table with the conceptual definition of each of them presented altogether. Ultimately, I think it's important to adopt some strategies to make it easier for the readers to follow and understand.

2. Some places need to be further elaborated with more evidence. For example, at the top of Page 3, you said, "All environmental designs can produce healing perceptions and have the potential factors to achieve healing effects." This claim needs to be explained with evidence.

3. Figure 1 should be revised for clarity. I found the mediating paths demonstrated (H6-H8) confusing and difficult to understand. Please also see my comment#1 for revision suggestions. You may also consider adding notes under the caption to clarify all the paths (e.g., solid lines vs. dashed lines).

4. For the Methodology – did you compensate the participants? How did you approach them? Was there any consent procedure involved? How many people have you approached in total, and how many refused your requests? Was the survey written in Mandarin Chinese or English? If in Mandarin Chinese, what were the translation and back-translation procedures? I'd like to see more details reported.

5. For the measurements - please either include all the items either as an Appendix, or at least include sample items, given that you conducted an SEM in this study.

6. For the Results – did you control for any of the demographic variables in the SEM model? Why or why not?

7. The Discuss section should be enhanced with more implications of this study. What are the theoretical and empirical contributions of this study? I'd like to see more specific potential applications of your study results in real-world situations, such as in policy recommendations for urban development, mental wellness programs using public parks, etc.

I hope the feedback will be helpful for the authors to strengthen this paper. 

Author Response

Response to reviewer 1

The authors greatly appreciate the kind comments and constructive suggestions. The responses to the reviewers' comments are addressed as follows one by one.

This is a very interesting study. I appreciate the authors' efforts in conducting this research with valuable implications. The manuscript was well-written, and the research was nicely executed.

R.: Thank you for your appreciation!

The structure of the front-end could be further improved for more clarity. Since you have included several sub-dimensions underneath each of the four main constructs: environmental design, healing perception, place attachment, and loyalty, it would be better to illustrate all the sub-dimensions either in the conceptual model (Figure 1) or create a table with the conceptual definition of each of them presented altogether. Ultimately, I think it's important to adopt some strategies to make it easier for the readers to follow and understand.

R.: We added the sub-dimensions under the four constructs in the figure 1 of the research model. [Figure 1, p. 6]

Some places need to be further elaborated with more evidence. For example, at the top of Page 3, you said, "All environmental designs can produce healing perceptions and have the potential factors to achieve healing effects." This claim needs to be explained with evidence.

R.: We eliminated the misleading word "All" in the sentence "All environmental designs can…" and added three citations for evidence as follows [Citations 23-25, line 98, p. 3]:

Zadeh, R.S., et al., Environmental design for end-of-life care: An integrative review on improving the quality of life and managing symptoms for patients in institutional settings. Journal of pain and symptom management, 2018. 55(3): p. 1018-1034.

Day, C., Places of the soul: Architecture and environmental design as a healing art. 2017: Routledge.

Schweitzer, M., L. Gilpin, and S. Frampton, Healing spaces: elements of environmental design that make an impact on health. Journal of Alternative & Complementary Medicine, 2004. 10 (Supplement 1): p. S-71-S-83.

Figure 1 should be revised for clarity. I found the mediating paths demonstrated (H6-H8) confusing and difficult to understand. Please also see my comment#1 for revision suggestions. You may also consider adding notes under the caption to clarify all the paths (e.g., solid lines vs. dashed lines).

R.: Thank you for pointing this out. In figure 1, the solid line indicates direct effect; while the dashed lines indicate indirect effects. We added  a legend to denote the meanings of the solid and the dashed lines. [Figure 1, p. 6]

For the Methodology – did you compensate the participants? How did you approach them? Was there any consent procedure involved? How many people have you approached in total, and how many refused your requests? Was the survey written in Mandarin Chinese or English? If in Mandarin Chinese, what were the translation and back-translation procedures? I'd like to see more details reported.

R.: In several two-person groups, the survey team went to the activity and sitting areas in the park to approach the potential participants. The surveyors introduced themselves, explained the purpose of the research, and requested participation in the questionnaire. A souvenir would be given after the participants completed the questionnaire designed for this study and presented in Mandarin Chinese for local participants' self-reporting. The visitors in the park were friendly toward the research, which is unintrusive to their privacy, and were willing to share their attitudes and feelings. The recruiting of participants was smooth. The survey team collected 449 responses from the 450 questionnaires distributed. 18 uncompleted responses were eliminated, and the remaining valid responses were 431.

Under this circumstance, we adjoined sentences and paragraphs the appropriate places in the paper as follows:

"The measurement items are originally written in Chinese." [line 308, p. 8]

"In several two-person groups, the survey team went to the activity and sitting areas in the park to approach the potential participants. The surveyors introduced themselves, explained the purpose of the research, and requested participation in the questionnaire. A souvenir would be given after the participants completed the questionnaire designed for this study and presented in Mandarin Chinese for local participants' self-reporting. The visitors in the park were friendly toward the research, which is unintrusive to their privacy, and were willing to share their attitudes and feelings. The recruiting of participants was smooth." [line 289-296,  page 8]

Moreover, we also revised the sentence as follows:

"The survey team collected 449 responses from the 450 questionnaires distributed. 18 uncompleted responses were eliminated, and the remaining valid responses were 431." [line 338-340, p. 9]

For the measurements - please either include all the items either as an Appendix, or at least include sample items, given that you conducted an SEM in this study.

R.: As suggested by the reviewer, we included the Appendices for the tables of the measurement items in the revision. [p. 18-21]

Moreover, we also cited the appendices in the revision as follows:

Appendix A [line 302, p. 8]

Appendix B [line 304, p. 8]

Appendix C [line 306, p. 8]

Appendix D [line 308, p. 8]

For the Results – did you control for any of the demographic variables in the SEM model? Why or why not?

R.: We did not include the demographic variables in the SEM model because of their weak impact on the constructs measured. The same condition would be expected in the model. Thus, the demographic variables were excluded from the model.

The Discuss section should be enhanced with more implications of this study. What are the theoretical and empirical contributions of this study? I'd like to see more specific potential applications of your study results in real-world situations, such as in policy recommendations for urban development, mental wellness programs using public parks, etc.

R.: We enhance the discussion to address the implications of the study results in practice.   

"This research demonstrated the crucial role of environmental design through psychological mechanisms, such as healing perception and place attachment, to influence the visitors to stay longer and often come in the park to benefit from abiding there." [line 439-441,  p. 12-13]

"In the practice, the planners and designers can utilize the relationship between environmental design, healing perception, and place attachment to create a cozy outdoor environment that improves human well-being. This principle is not only valid for the urban park, but should be promoted in any place which longs to have a healing effect on the visitors." [line 456-460, p. 13]

"All of the six indicators of environmental design in the model retain high factor loadings and can be employed as the key design factors to conceive a healing environment." [line 460-462, p. 13]  

"The mechanism that the study revealed can be used in the design and planning sector to create places with humanity and healing effect and should be introduced to the health care and well-being sector to enhance the healing perception by abiding in an appropriately designed public park." [line 477-481, p. 13]

I hope the feedback will be helpful for the authors to strengthen this paper.

R.: Thank you so much for the comprehensive feedback. Your suggestions are very productive in strengthening our paper. The authors are grateful deeply.

Reviewer 2 Report

This article uses structural equation modeling (SEM) in order to analyze the relationships between healing perception, place attachment, environmental design, and visitors’ loyalty to a place, specifically in the case of a Gaoxiong Weiwuying Metropolitan Park (Taiwan). It theoretically relies on Ulrich and Kaplan's separate studies, and extracts from them some common strategies for healing mental health thourgh healing perceptions by means of environmental design. Then it explores the idea of 'Loyalty' in terms of place attachment throughout the time. The method includes data collection through questionnaires and results are provided through the Structural Equation Modeling (SEM). 

It provides an interesting overview of relations and works that have  been carried out so far, and frames properly the study. The diagram referred to as 'Figure 1' is particularly remarkable. 

As a suggestion, there is a whole branch from human geography focused on the idea of place. Specially remarkable were Edward Relph and Yi-Fu Tuan's works in the 70s and 80s, as well as Christian Norberg-Schulz's. Authors may include a reference to how deep are the concepts of place and place attachment and a little bit of its history. 

The article could be a little bit shorter, some explanations are longer than necessary. When language copyediting is made, this could improve substantially. 

Minor revisions are suggested, basically related to English copyediting and simplifying sentences. 

Author Response

Response to reviewer 2

The authors greatly appreciate the kind comments and constructive suggestions. The responses to the reviewers' comments are addressed as follows one by one.

1.This article uses structural equation modeling (SEM) in order to analyze the relationships between healing perception, place attachment, environmental design, and visitors' loyalty to a place, specifically in the case of a Gaoxiong Weiwuying Metropolitan Park (Taiwan). It theoretically relies on Ulrich and Kaplan's separate studies, and extracts from them some common strategies for healing mental health through healing perceptions by means of environmental design. Then it explores the idea of 'Loyalty' in terms of place attachment throughout the time. The method includes data collection through questionnaires and results are provided through the Structural Equation Modeling (SEM).

It provides an interesting overview of relations and works that have been carried out so far, and frames properly the study. The diagram referred to as 'Figure 1' is particularly remarkable.

R.: Thank you for your appriciation.

2.As a suggestion, there is a whole branch from human geography focused on the idea of place. Specially remarkable were Edward Relph and Yi-Fu Tuan's works in the 70s and 80s, as well as Christian Norberg-Schulz's. Authors may include a reference to how deep are the concepts of place and place attachment and a little bit of its history.

R.: Thanks for your kind suggestion. We provide a short history of human geography related to the idea of  “place” and 3 new relevant citations [27-29]. [line 105-107, p. 3]

"In the 70s and 80s, the branch of human geography mentioned the idea of place, which profoundly impacted the concept of place and place attachment [27-29]."

Tuan, Y.-F., Space and place: humanistic perspective, in Philosophy in geography. 1979, Springer. p. 387-427.

Norberg-Schulz, C. (2013). The phenomenon of place. In The urban design reader (pp. 292-304). Routledge.

Relph, E., Sense of place. Ten geographic ideas that changed the world, 1997: p. 205-226.

3.The article could be a little bit shorter, some explanations are longer than necessary. When language copyediting is made, this could improve substantially.

R.: Thanks for your kind reminder.

4.Minor revisions are suggested, basically related to English copyediting and simplifying sentences.

R.: Thank you for the kind suggestion. We went through the entire manuscript for the English copyediting and simplifying the sentences.
